# Motivating Individuals to Take Responsible Ocean Action: The Mediatory Effects of Attitude toward the Ocean

**DOI:** 10.3390/ijerph20032676

**Published:** 2023-02-02

**Authors:** Guang-Ying Liu, Yi-Chen Lin, Ting-Kuang Yeh

**Affiliations:** 1Department of Earth Science, National Taiwan Normal University, Taipei 116059, Taiwan; 2Institute of Marine Environment Science and Technology, National Taiwan Normal University, Taipei 116059, Taiwan

**Keywords:** ocean literacy, attitude toward the ocean, marine education, mediatory effects

## Abstract

When considering how to improve public literacy and behavior related to specific themes, top priority is usually given to strategies that enhance relevant knowledge. Fostering attitude comes later. Understanding the mechanisms of behavior may help us develop better policy and educational strategies. However, how knowledge and attitude impact behavior is still under investigation. The aim of this study is to explore the relationships among ocean knowledge, attitude toward the ocean, and the intention to behave responsibly in the marine setting. Specifically, we investigated a potential mediation mechanism by recruiting a total of 266 participants, whose ocean knowledge, attitudes toward the ocean, and intention to behave responsibly were evaluated using questionnaires. The results indicate that a person’s attitude toward the ocean may indeed be a mediating factor between ocean knowledge and their intention to show positive marine behavior. In order to engage people in responsible ocean behavior, other forms of assistance from marine policy and education are recommended. Additionally, it would be of interest for future studies to investigate the effects of attitude and attitude-related knowledge in the development of ocean actions.

## 1. Introduction

The oceans cover over 70% of the Earth’s surface and provide crucial social, economic, and environmental benefits to the earth’s growing population. According to the OECD’s 2017 Green Growth and Sustainable Development Forum on “Greening the Ocean Economy,” the oceans support activity with an annual economic value in the range of USD 1.5–2 trillion. In 2017, the OECD estimated that the value of the ocean economy might more than double by 2030 [1]. However, marine resources are in rapid decline due to the destruction of habitats, overfishing, pollution, and global change. The promotion of ocean literacy and encouraging citizens to behave in positive ways toward the ocean have been regarded as challenging yet fundamental elements of minimizing the negative environmental impacts caused by human activity [2,3]. In the pursuit of this goal, a series of ongoing ocean education research projects at international and national levels has been proposed within the Sea Grant Program (US National Oceanic and Atmospheric Administration, https://seagrant.noaa.gov/, accessed on 20 December 2022), the EU Blue Growth Program (European Commission, https://blue-action.eu/policy-feed/blue-growth, accessed on 20 December 2022), and the Ocean Literacy Program (Japan, https://www.cole.p.u-tokyo.ac.jp/en/education, accessed on 20 December 2022), among others.

Current developments in scientific education have placed a strong emphasis on scientific literacy, which has now emerged as a clear common theme [4,5,6]. Marine education is presently evolving along the same path as marine educators work toward the goal of integrating ocean literacy into classroom activities. Ocean literacy is a broad concept that encompasses a number of educational themes that have shifted over time. Many institutions and individuals have tried to define what is meant by ocean literacy. As part of a project supported by the National Geographic Society, the Centers for Ocean Sciences Education Excellence, the National Marine Educators Association (NMEA), the US National Oceanic and Atmospheric Administration, and the College of Exploration, Cava, Schoedinger, Strang, and Tuddenham [7] developed a framework for the Essential Principles and Fundamental Concepts of Ocean Sciences. In their report, ocean literacy is defined as a means to “understand the essential concept, communicate about the ocean in a meaningful way, as well as make informed and responsible decisions regarding the ocean”.

Following the development of initial principles and concepts with respect to ocean literacy, some adjustment is still ongoing due to the wide range of meanings and the variety of goals related to ocean literacy. The development of ocean literacy is similar to the development of scientific literacy due to the plethora of visions of scientific concepts, processes, the nature of science, procedural skills, and affective and motivational development, both for general and local education purposes, all of which should be included in the notion of scientific literacy [6,8]. In the public realm, teachers and researchers generally acknowledge that scientific literacy encompasses cognitive, affective, and practice domains (NGSS). Therefore, in line with scientific literacy, ocean literacy should also include, albeit not exclusively, marine conceptual understanding, procedural skills, and affective and motivational development, with its ultimate aim being the development of responsible behavior regarding the ocean [8].

Given that human-induced modifications have been the dominant detectable threat to the global marine environment [9], understanding cognitive and affective mechanisms of responsible behavior regarding the ocean may aid agencies in refining policies and procedures that can enhance the sustainable development of the oceans. In recent years, there has been a growing interest in studies of ocean literacy, with the primary focus on investigating the level of knowledge of ocean science [2,10]. However, even though a number of studies have probed the relationship between ocean knowledge and behavior, their results are not consistent. While many studies have shown that knowledge significantly influences environmental actions, others have indicated the poor correlation between technical know-how and behavior.

According to Steel, Smith, Opsommer, Curiel, and Warner-Steel [11], improving citizens’ knowledge should be the first step in establishing a nationwide effort to preserve the oceans in the USA. Moreover, McKinley and Fletcher [12] identified lack of knowledge as a major factor in public non-involvement in environmental activities. Likewise, Hines, Hungerford, and Tomera [13] argued that knowledge is a prerequisite of action in this regard. In terms of recent studies, Umuhire and Fang [14] highlighted that a lack of knowledge can limit participation in ocean-related action. On the other hand, Meinhold and Malkus [15] examined the relationships among adolescent behaviors, knowledge, and attitudes and found attitudes to be better predictors of behaviors than knowledge. A similar finding was reported by Pe’er, Goldman, and Yavetz [16] for 765 students in Israel, where attitudes correlated with environmental behavior more strongly than knowledge. Though academics agree on the importance of knowledge in environmental actions, they are of divergent opinions concerning whether attitude contributes a larger impact on behaviors than knowledge [17]. Nonetheless, there is another possible mechanism, namely, the effect of knowledge on behaviors may be mediated by other factors [18]. According to Lin et al. [19], a mediation mechanism can be conjectured as the reason for an indefinite behavioral pattern due to the consideration of multivariable interactions. Aligned with this idea, Ajzen [20] firstly found that the relationship between knowledge and behavior appeared weak. With further exploration, Ajzen, Joyce, Sheikh, and Cote [21] investigated the effect of knowledge on behaviors in varied behavioral contexts, which showed that mediation mechanisms might exist in specific behaviors, navigating the intention of actions through the interaction of knowledge, attitude, and behaviors. This finding revealed that behavioral features might differentiate the behavioral models. In other words, the attitude–knowledge–behavior patterns might be distinctive in terms of behavioral characteristics. Despite protracted discussion on the possible mediation effect within behavioral mechanisms, little work has been done to provide empirical evidence on the mediation mechanism within marine environmental behavior. The aim of this study is to fill this gap by constructing a mediation model. We hypothesize that attitude is a mediator that lessens the impact of knowledge. This assumption has emerged from interpretations of the statistical results of aforementioned studies. In terms of regression analysis, some studies revealed knowledge to be powerless in terms of influencing behavior. However, if the argument that knowledge contributes to behavior indirectly is tenable, knowledge will impose its impact on the mediator—attitude; consequently, attitude will significantly influence behavior, whereas the effect of knowledge will vanish. On the basis of the hypothesized mediation effect, we present the specific purpose and methodology of this study in the following sections.

Furthermore, it is possible that we may fail to estimate the impact of knowledge on behavior if the population has a general lack of basic knowledge. Indeed, this could be one of the reasons for the inconsistent results of the knowledge–attitude–behavior studies mentioned above. Taiwan is one of only a few nations to have promoted marine education in the national curriculum. Therefore, conducting research in the Taiwanese population could help better understand the correlations among knowledge, attitude, and behavior.

### Purpose of the Study

Characterization of the ocean literacy of citizens is the first stage in longitudinal, cross-sectional, and causal research that forms part of the investigation of the relevant positive behavioral mechanisms. Information gained from this study is vital in view of its potential for promoting marine policy and the development of marine education programs. The aim of this article is to examine a model of intentions toward responsible marine behavior via the characterization of ocean literacy, including ocean knowledge, attitudes toward the ocean, and intentions toward responsible marine behavior.

## 2. Methodology

### 2.1. Participants

As mentioned previously, Taiwan is one of only a few nations to have promoted marine education in the national curriculum. For the purpose of fostering ocean literacy among its citizens, the Ministry of Education has mandated that marine education be included in the national curriculum guidelines for secondary education since 2008 [22]. This means that all citizens under 24 years old in Taiwan received basic ocean education at middle school. Therefore, those aged less than 15 or more than 24 were excluded from the present study. A total of 266 participants, comprising 136 males (51%) and 130 females (49%), were recruited for the study. This included 184 high school students (69%) and 82 college students (31%), who were selected from three major geographical regions in northern, central, and southern Taiwan. All participants provided informed consent.

### 2.2. Evaluation of Scientific Literacy

To measure subjects’ ocean literacy in terms of their conceptual understanding, their attitudes toward the ocean, and their intentions regarding responsible marine environmental behavior, we devised and developed the Ocean Conception Instrument (OCI) and employed the Attitudes towards the Ocean Inventory (AOI) and the Intention to Take Responsible Marine Action Instrument (ITRMAI). These are explained in more detail below, and sample questions are shown in Table 1.

### 2.3. Ocean Conception Instrument (OCI)

The OCI is a 75-question multiple-choice test designed to measure subjects’ conceptions of the ocean. Given that the subjects in this study are Taiwanese students, the development of the OCI is mainly based on the topics highlighted by the Marine Education Curriculum Guidelines of Taiwan [22], as shown in Table 2. The Marine Education Curriculum Guidelines of Taiwan comprise marine leisure, marine society and culture, marine science and technology, and marine resources and sustainable development topics. Though Taiwanese guidelines encompass diverse marine conceptions, the great breadth of content could render obstacles in the selection of ideal questions. For example, the major aim of marine leisure is to develop skills and the motivation to do outdoor marine activities, rather than learn about concepts. In this regard, marine leisure may not be conceived as ideal knowledge for test content. In order to address this issue, we took the Ocean Literacy definition informed by NOAA [23] as it provides essential principles and fundamental concepts of ocean science for learners of all ages. Specifically, we designed OCI test items by prioritizing the content that overlaps with NOAA’s Ocean Literacy. Complemented by NOAA’s standard, OCI items are considered more likely to assess subjects’ marine concepts. With respect to the validity and reliability of the OCI, the content validity was determined through a panel of six marine science and marine education professionals, instructors, and graduate students. The reliability of the scale was assessed by KR20, which yielded a value of 0.85.

### 2.4. Attitudes towards the Ocean Inventory (AOI)

Research on attitudes related to science and education has mostly concerned perceptions, values, self-beliefs, interests, and motivations concerning science and education [24,25,26,27,28]. In order to measure subjects’ attitudes toward the ocean, we developed the AOI, which was revised from the PISA (Program for International Student Assessment) science attitude scale [29]. The AOI is a five-point Likert-type scale that consists of 26 items that are divided into five categories: Support for Marine Science (items 1–6), Self-Confidence in Learning about the Ocean (items 7–13), Interest in Learning More about the Ocean (items 14–17), Motivation for Learning about the Ocean (items 18–22), and Responsibility towards Ocean Resources and Environments (items 23–26). The reliability of the AOI was assessed using Cronbach’s alpha, which yielded a coefficient of 0.91. Construct validity was examined through confirmatory factor analysis (CFA). The CFA confirmed the scale with its five factors (Figure 1).

### 2.5. Intention to Take Responsible Marine Action Instrument (ITRMAI)

The Intention to Take Responsible Marine Action Instrument comprises 10 items used to assess the extent to which individuals are willing to engage in actions to protect the ocean environment (eco-management, consumerism, persuasion, and civic actions). These items were generated from the literature [30,31,32,33] using a five-point Likert-type scale. The Cronbach’s alpha coefficient of the scale was 0.88.

### 2.6. CFA

To establish the validity of the AOI questionnaire, we conducted confirmatory factor analysis (CFA) using AMOS to re-establish whether attitude can be constructed by measuring five indicators (as shown in Figure 1), namely perception, value, self-belief, interest, and motivation. Our standard of factor loadings was 0.5. The criteria listed below were used to assess the model fit: the value of *𝜒*^2^/*df* must be less than 3.0 [34]; the comparative fit index (CFI) must be more than 0.90 [35]; the root-mean-square error of approximation (RMSEA) must be less than 0.08 [36,37].

### 2.7. Tests for Mediation

In recent years, the bootstrapping approach has replaced the causal steps approach [38] as a more mainstream method for testing mediation hypotheses [39,40,41]. To examine the hypothetical model presented in Figure 2, the SPSS PROCESS macro was utilized to carry out a path analysis [42].

## 3. Results

### 3.1. The Results of the OCI

Prior to the evaluation of the subjects’ understanding of ocean literacy, the internal consistency of the OCI items was established using the Cronbach’s *α* coefficient (*α* = 0.86). The results of the descriptive analysis showed a mean score for the entire sample of 45.05 (*SD* = 10.25). Our data also indicate that the following principles had the greatest level of misconception: the major mechanism of global sea level rise (passing score: 12%); the reason why Kuroshio is known as the black current (28%); the major energy source for typhoons is warm ocean water (29%); whale sharks are fish, rather than mammals such as whales (30%); the classification of ocean animals (34%); and tiny white fish (whitebait) consist of immature fish from a number of different species (35%).

### 3.2. Mediation Test via a Path Model

Firstly, a confirmatory factor analysis was undertaken to identify an attitude model that consisted of five indicators (perception, value, self-belief, interest, and motivation). The results revealed a model with an adequate fit to the data (*𝜒*^2^ = 532.70; *df* = 289; *p* < 0.001; *𝜒*^2^/*df* = 1.84; CFI = 0.94; RMSEA = 0.06). Specifically, all values met the criteria for acceptable fit. The value of *𝜒*^2^/*df* was less than 3.0 [34], the comparative fit index (CFI) was more than 0.90 [35], and the root-mean-square error of the approximation (RMSEA) was less than 0.08 [36].

Second, a process macro output for the mediation test was obtained as presented in Table 3. The results suggest that attitude is a significant mediator between ocean knowledge and the intention to take responsible marine action (ITRMA). The coefficient of indirect effect along the a path (knowledge–attitude) was 0.013 with *p* < 0.001, while for the b path (attitude–ITRMA) it was 1.13 with *p* < 0.001. The direct effect of the relationship between knowledge and ITRMA (c′-path) was insignificant. The total effect of knowledge on ITRMA, mediated by attitude, was *R*^2^ = 0.59 and *F* (2, 263) = 191.15 with *p* < 0.001.

## 4. Discussion

### 4.1. Subjects’ Conceptions of Ocean Literacy

To characterize ocean literacy in terms of knowledge, the OCI questionnaire was undertaken to explicitly evaluate seven essential principles regarding the ocean. The results reveal that even though Taiwanese subjects received education about the ocean at an early stage, the conceptual knowledge of the ocean that they exhibited is nevertheless flawed and fragmentary. One striking example is the mechanism relating to storm surges. Despite the fact that Taiwan is located in a subtropical zone affected by storm surges, and that global warming has dramatically increased the impact of storms on Taiwan, the subjects’ conceptions were still inconsistent with those of teachers and scientists.

The data also reveal that the subjects struggled to comprehend long-term abstract concepts that cannot be observed directly, such as global change, the principles and phenomena of the ocean, and ocean environmental issues. This finding is similar to those from a previous study on misconceptions in environmental and earth science education [43,44,45]. The learners often experience cognitive overload in learning, develop misconceptions, and cannot integrate this knowledge into their behavior in daily life [46]. To foster learners’ conceptual understanding of ocean literacy, their learning must entail integration of multiple displays of data, such as graphs, models, and other figures. In this regard, new teaching materials and strategies are required to meet the learning needs of the students. For instance, students usually face a considerable challenge in learning ocean environmental issues due to their long temporal and/or large spatial scales.

### 4.2. Mediation Effect of Attitude between Knowledge and Intention to Take Responsible Marine Action (ITRMA)

When a discussion touches on responsible environmental behavior, attention is often paid to two dominant opinions. While many believe that knowledge directly influences behavior [11,12,13,14], others are convinced that behavior is more susceptible to attitude than knowledge [15,16,47,48,49]. However, evidence to explicate the pattern of how knowledge and attitude contribute to behavior is still lacking. Since these ideologies may influence the policy-making process and the development of the education system, it is prudent and necessary to explore them impartially before jumping to arbitrary conclusions. In this regard, the present study developed a path model based on the potential mediation effect.

As hypothesized, attitude is verified as a significant mediator between knowledge and behavior. In other words, the current study provides statistical evidence that the impact of knowledge on behavior is not direct [50]; rather, it indirectly influences behavior though knowledge-related attitude. The findings of a meta-analysis of pro-environmental behavior conducted by Bamberg et al. [50] support this pattern. Their data lend support to the fact that knowledge is an indirect precondition for pro-environmental behavior. The cognitive preconditions, such as awareness and knowledge, develop the attitude and moral norm of subjects and turn into the foundation for their choice of pro-environmental behavior. This finding is also aligned with the study by Ajzen et al. [21], which compared the correlation between knowledge and behavior in several contexts.

The benefits of setting up an education system with an emphasis on knowledge are traceable. A well-developed category of marine conceptions provides an explicit framework for authorities to structure formal environmental curriculums. However, even though several countries are committed to promoting environmental behavior such as energy saving and waste resource reduction, investigators have recently found that filling public knowledge deficits does not engage citizens in greater actions [51]. Not only have government organizations experienced such frustration, but the “gray literature” produced by NGOs indicates that only few people are resolved to practice desirable behaviors, despite the fact that some of them are highly intellectual (Goodwin Simon Strategic Research and Wonder: Strategies for Good, https://www.packard.org/wp-content/uploads/2019/05/Heartwired-to-Love-the-Ocean-Final.pdf, accessed on 20 December 2022). To address this issue, there is a growing body of research that advocates taking attitude into consideration [15,16]. Apart from these claims, our finding of a mediation effect provides another perspective, whereby the interpretation of the correlation between knowledge and attitude has to be carefully thought out.

Given that the extent of marine conceptions indirectly shapes environmental behavior through the contribution of attitude, the development of behavior necessitates policy and an education system that can combine knowledge communication and attitudinal identification. Another interesting question concerns the types of conceptions and the kinds of instructions that impact learners’ attitudes. These questions nonetheless hold some noteworthy implications for marine education policy and represent a significant challenge for the future. In terms of the promotion of students’ attitudes, previous work has shown the non-traditional pedagogical approach may contribute to better attainment in terms of learning interest than traditional classroom settings as it offers unique and motivating experiences. Outdoor learning is a case in point. With field trips and hands-on experiences, students are found to be more likely to better understand ocean literacy and maintain their curiosity through lifelong learning [52,53,54]. Additionally, other educational approaches have also been shown to be effective in enhancing students’ attitudes. For instance, increasing evidence suggests that information related to animals suffering from human actions will induce pro-environmental behavior, mainly because the conception may stimulate empathy toward animals [55,56,57]. Issue-based learning or experimental learning cycles may be an effective approach as they have been shown to cultivate students’ attitudes through conceptualization and reflection activities. Game-based approaches may also be helpful. Goldberg et al. [58] indicated that our attitudes formed via deep engagement with game-based arguments are “often more consistently organized around core principles in one’s belief system”. Another approach is dilemma-based instruction activities. Learners may have enduring attitude changes as they practice defending their position and persuading the opposing party in this kind of instruction. If they have the chance to practice defending their beliefs, they will more likely gain immunity to fake information from social media in the future [59]. Finally, since many environmental changes cannot be observed directly, visualization techniques are also recommended when they offer a realistic representation of the processes to be explained, facilitating a deeper comprehension of dynamic systems and providing a greater motivation to learn about them [60,61].

## 5. Conclusions

The majority of previous studies have been dedicated to comparing the explanatory power of knowledge and attitude toward behavior. Consequently, many studies have neglected the importance of knowledge; indeed, a relatively low correlation between knowledge and behavior was found in this study. However, the results for the mediating effect demonstrate the potential benefits of disseminating knowledge and its strong relationship with attitude. In addition to the promotion of engagement, environmental knowledge may underlie the quality of a person’s behavior. Establishing adequate knowledge is imperative to the adjustment of behavior. Our argument greatly resembles that of Goldberg et al. [58]. The present study reiterates the focus on both cognitive approaches and affective approaches in terms of fostering behavior, which are indispensable and are inherently interdependent. Attitude can provide visceral motivation to people who only acknowledge environmental problems but do not generate positive behaviors. On the other hand, “people may feel negative emotions toward climate change, but without specific knowledge about solutions, people will be less likely to connect their feelings to concrete action”.

## Figures and Tables

**Figure 1 ijerph-20-02676-f001:**
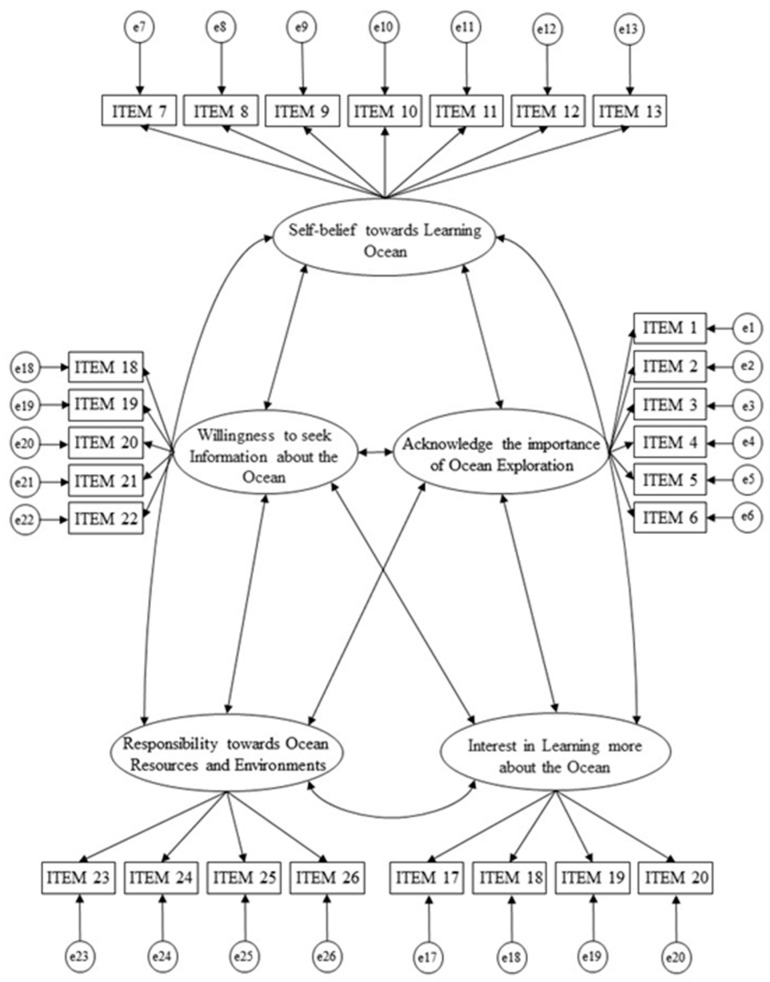
The CFA-hypothesized model of the AOI on AMOS.

**Figure 2 ijerph-20-02676-f002:**
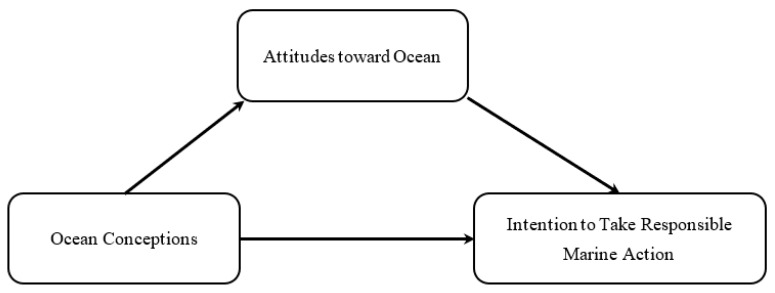
The hypothesized mediation model.

**Table 1 ijerph-20-02676-t001:** AOI, ITRMAI, and OCI examples.

Instruments	Constructs	Examples
AOI	Acknowledge the Importance of Ocean Exploration (Values)	It is essential for the government to provide an annual budget for researchers to explore the ocean. Ocean exploration is important because it helps us understand the natural environment around us.
	Self-Confidence in Learning about the Ocean (Self-Belief)	I can easily understand the ocean-related knowledge provided in school courses. If there are some ocean-related concepts that I don’t understand, I can always collect relevant data and perform analyses to figure out the right answer.
	Interest in the Ocean (Interest)	I am always delighted to discover new knowledge about marine matters. I am interested in outdoor experiments related to marine matters.
	Willingness to Seek Information about the Ocean (Motivation)	I find the ocean-related course provided at this stage very helpful, because it will help me greatly with my future career. I hope to pursue an ocean-related career.
	Responsibility toward Ocean Resources and Environments (Perception)	Marine ecology is closely related to my daily life. It is our duty to protect the beach and seashore.
ITRMAI		I would like to participate in coastal clean-up activities. I would like to take time to think about how to maintain the sustainable development of the ocean.
OCI	Marine Physics	What is the main factor of sea level rise? (A) Melting of sea ice (B) Thermal expansion of seawater (C) Mid-ocean ridge expansion (D) El Nino phenomenon
	Climate Change and Sustainable Development	The primary cause of tides is the lunar gravitational force, followed by the solar gravitational force. With regards to tides on Earth, what is the main reason why the lunar gravitational force is stronger than the solar gravitational force? (A) The moon has a greater mass than the sun. (B) The moon is closer to Earth than the sun. (C) The moon always faces the same side towards Earth. (D) The moon orbits and rotates around Earth at the same time.

**Table 2 ijerph-20-02676-t002:** Item distribution in accordance with components of the marine education curriculum of Taiwan.

Constructs	Sub-Constructs	Items
Marine Culture and Society	Marine Culture and Society	8
Marine Science and Technology	Marine Physics	12
	Marine Chemistry	11
	Marine Geology and Geography	13
	Marine Meteorology	17
Marine Resources and Sustainable Development	Marine Biology	10
	Climate Change and Sustainable Development	9

**Table 3 ijerph-20-02676-t003:** Results of mediation tests via process macros. ITRMA: Intention to Take Responsible Marine Action.

Consequent								
	Attitude				ITRMA			
Antecedent	Path	Coeff.	SE	*p*	Path	Coeff.	SE	*p*
Knowledge	a	0.013	0.003	<0.001	c′	−0.001	0.003	0.818
Attitude					b	1.132	0.060	<0.001
Constant	i1	2.973	0.136	<0.001	i2	−0.646	0.222	<0.01

## Data Availability

The data presented in this study are available on request from the corresponding author.

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
