# Peer review of "Motivating Individuals to Take Responsible Ocean Action: The Mediatory Effects of Attitude toward the Ocean"

_ijerph, 2023, doi:10.3390/ijerph20032676_

Round 1

Reviewer 1 Report

The current manuscript is ready to be published after minor revision of the following:

1.     Renew some of the relatively older literature.

2.     Advice with English native speaker editing service.

3.     Include full survey question items.

“Attitude” in this study as a significant mediator between knowledge and behavior. In other words, the current study provides statistical evidence that the impact of knowledge on behavior is not direct”.

However some of the cited literature could be updated with more recent ones (As highlighted in the attached version of the manuscript).

It would be nice to include the full survey question items (table 1)

Author Response

Our thanks, to the reviewer for the insightful comments made. We have tried our best to revise this paper based on the reviewer's kind suggestions.

Reviewer 2 Report

Journal: International Journal of Environmental Research and Public Health

Manuscript ID: ijerph-2164588

Title: Motivating Individuals to Take Responsible Ocean Action: The Mediatory Effects of Attitude toward the Ocean

It is suggested that authors check or adapt what are referred to as objectives in the abstract and at the end of the introduction.

More detailed information on the questionnaire used as well as its results should be provided.

Bibliographical references are updated, corrected, checked for adequacy taking into account what has been published. See also some notes that it is suggested the authors may consider if they deem relevant. Please see, for more details, the pdf document.

Author Response

(The authors gave the same response as above.)

Reviewer 3 Report

This study investigated the relationship between attitude toward the ocean, ocean knowledge, and marine behavior among 15-24-year-old participants in Taiwan. This is a very interesting topic and a relationship worth understanding. After reading this manuscript, the following are some comments for the authors.

1. A general view of the relationship between attitude toward the ocean, ocean knowledge, and marine behavior is missing.

2. The instruments in this paper were developed by the authors. Did the authors conduct a pilot study on the scales? What are the reliability and validity of these scales?

3. The OCI is based on the seven principles of ocean literacy and five topics of the Marine Education Curriculum Guidelines of Taiwan, so how many components does the OCI of this study contain?

4. Continuing from the previous question, the OCI contains a total of 75 items, which should be detailed to show how the items are distributed in each component.

5. Describe more about the background of the participants, e.g., age distribution? percentage of students?

6. What are the analysis results of the CFA hypothesized model?

Author Response

(The authors gave the same response as above.)

Round 2

Reviewer 3 Report

The authors completed their responses according to the reviewers' comments.